# Engineering Integrated Care Expansion and Innovation: Drawing upon Nursing Leadership

**DOI:** 10.3390/ijerph22040598

**Published:** 2025-04-11

**Authors:** Kathleen R. Delaney, Margaret R. Emerson, Victoria Soltis-Jarrett, Amy J. Barton, Mary Weber

**Affiliations:** 1College of Nursing, Rush University Medical Center, Chicago, IL 60201, USA; 2College of Nursing, University of Nebraska Medical Center, Omaha, NE 68588, USA; margaret.emerson@unmc.edu; 3School of Nursing, University of North Carolina at Chapel Hill, Chapel Hill, NC 27599, USA; vsoltis@email.unc.edu; 4College of Nursing, University of Colorado Anschutz Medical Campus, Aurora, CO 80045, USA; amy.barton@cuanschutz.edu (A.J.B.); mary.weber@cuanschutz.edu (M.W.)

**Keywords:** integrated behavioral health care, psychiatric nurse practitioner, nurse-led care, collaborative care model

## Abstract

In the United States (US), a longstanding solution to the unmet need for mental health treatment is integrated behavioral health care (IBH). Within a primary care model, problems are identified and treatment combines care for physical, mental and substance use disorders. Treatments are delivered through the collaboration of primary and behavioral health providers. According to US federal billing guidelines, in one integrated model, the Collaborative Care Model (CoCM), the psychiatric consultant must be a medical professional trained in psychiatry and capable of prescribing medications, i.e., either a psychiatrist, Psychiatric Mental Health Nurse Practitioner (PMHNP) or Physician Assistant. The development of integrated care has been slow for particular vulnerable populations, in part due to the lack of psychiatric consultants. PMHNPs are increasingly taking on the role of psychiatric consultants on CoCM teams and creating nurse-led IBH models for underserved populations. In this paper, eight such models are discussed along with implementation challenges and the strategies used to address them. Nurse leaders have the capacity to enhance and expand integrated care, particularly for underserved populations, through the optimal utilization of care teams, expanding measured outcomes, and developing measures for team-based effectiveness. Future directions are proposed that will accelerate this PMHNP-led expansion of IBH.

## 1. Introduction

The concept of integrated care is quite broad and includes service models where care processes are organized to achieve improved outcomes, ensure the efficiency of resources, and provide an enhanced patient experience [1]. For example, a key integration model, known as the chronic care model [2], facilitates collaboration between community resources and the health system in providing care for patients with specific chronic conditions while also supporting self-management, e.g., [3,4]. A notable example of integration can be found in behavioral health-integrated care where health problems are identified within a contemporary primary care model and treatments address physical, mental and substance use disorders [5]. This range of treatments is made possible through the partnership of primary and behavioral health providers. Various models of integrated behavioral health care exist including Collaborative Care (CoCM) and Primary Care Behavioral Health (PCBH) [6,7]. Integrated behavioral health care (IBH) is increasingly acknowledged as an effective strategy for meeting the unmet mental health needs in the United States (US) population [8,9]. Advocacy groups highlight that IBH matches consumer preference for accessing mental health services within the primary care system, reduces stigma, improves access and addresses the common medical co-morbities often experienced by individuals living with mental illness [10].

In reviewing the history of IBH, experts trace the early adoption of integrated care, from its beginning in various Federally Qualified Health Centers (FQHCs) to its acceleration in the 1990s, largely owing to significant policies acknowledging the importance of IBH, such as the 1999 Surgeon General’s Report on Mental Health and the Affordable Care Act, which introduced the concept of whole-person care (also known as person-centered care) [11]. Soltis-Jarrett’s [12] historical account of IBH also examines the impact of mental health being “carved out” of community health centers in the 1980s and how this hindered integrated care development. She highlights the substantial influence of the IMPACT model (Improving Mood-Promoting Access for Collaborative Treatment), particularly due to the rigorous data collection that supported its success in treating depression

Given IBH’s documented achievements in depression treatment [13], viewpoints differ on the progress of IBH in addressing broader mental health needs across particular populations and communities [14,15,16]. Maps of integrated care practices indicate significant differences in density depending on the US region (https://www.cfha.net/resources/what-is-integrated-care/) (accessed on 3 January 2025) and IBH clinics are particularly lacking in some areas of the country with the highest mental distress [17]. Progress has been slow in developing IBH for individuals with Serious Mental Illness (SMI), which demands addressing both unmet mental health needs and medical issues [18] and also creating protocols for common co-morbidities such as hypertension, dyslipidemia and diabetes [19].

One barrier to expanding integrated care is the insufficient number of primary care providers trained to address mental health concerns beyond anxiety and depression [20] as well as the shortage of psychiatric providers to take on the psychiatric consultant role [21]. The Health Resources and Services Administration (HRSA) projects that by 2030, there will be a shortage in the United States of 37,000 psychiatrists, the traditional providers of psychiatric consultation services in IBH models [21]. Allied professionals such as community health workers (CHWs) are also needed and essential for bridging clinical services with community level interventions [11]. Strategies to expand the IBH workforce include raising payment rates for CoCM, creating pipeline programs, broadening loan repayment options for clinicians, developing career paths for CHWs and increasing the number of providers capable of acting in the consultant role [22,23]. The consultant role in CoCM is filled by professionals whose education and licensure enable them to deliver psychiatric services, including physicians, nurse practitioners (NPs) and physician assistants (PAs) [24]. Research shows that NPs are vital to expanding mental health and substance use services, particularly in rural areas and among older adults [25,26,27]. Moreover, the ability to provide holistic care regardless of a nurse practitioner’s specialty training appears to be inherent in nursing, aligning perfectly with IBH treatment approaches [28].

While Psychiatric Mental Health Nurse Practitioners (PMHNPs) are well suited for integrated care roles, their training for these positions has progressed slowly [29]. As a result of workforce development funding opportunities through HRSA, there are efforts to embed interprofessional training for IBH within graduate programs, e.g., [30]. While interprofessional IBH training is evident in the literature [31], efforts to incorporate integrated care training into PMHNP graduate education are few [32,33,34]. The Bipartisan Policy Center’s review on training trends cites the lack of “real world” exposure to IBH during training, together with a focus on discipline specific training, which results in providers who are not able to function as part of an integrated care team [22]. Moreover, much of the literature on IBH and training for psychiatric consultant roles has focused primarily on psychiatrists or physicians rather than PMHNPs (https://aims.uw.edu/psychiatric-consultant/) (accessed on 5 January 2025). 

Despite these challenges, models of nurse-led CoCM and PMHNP participation in IBH development are increasingly reported in the literature. However, these programs are scattered and not examined as a collective of evolving models, which would synthesize their gains as well as identify implementation issues and the strategies to address them. In light of the shortage of providers to fill the psychiatric consultant position, this information would inform efforts to expand IBH with PMHNPs in this role [22].

The aim of this paper is to summarize the recent literature demonstrating the development and outcomes of nurse-led models of IBH with a focus on the utilization of a PMHNP as the psychiatric provider/consultant. An unstructured literature review was used to explore the research aim. The lead author, experienced in psychiatric nursing advanced practice models of care, reviewed the academic and gray literature with search the terms “PMH-NPs”, “integrated behavioral health care”, “collaborative care”, and “nurse-led models of care”. Models were included where a PMH-NP practiced as the psychiatric provider/collaborator/psychiatric consultant. Authors of several identified models were enlisted to enrich the understanding of their program development and challenges faced. We did not include integrated behavioral care models which were developed by nurses but did not utilize a PMHNP as the psychiatric provider.

These models are quite diverse and describe PMHNPs assuming the CoCM consultant role and developing new integration models that fit the population and service site. In line with the implementation framework, Normalization Process Theory (NPT), beyond a description of each model, we identified aspects of the models that were subject to known barriers to IBH implementation and the influence of context, particularly how context demanded specific strategies around resource mobilization, negotiations with the interdisciplinary team and modifications to standard CoCM processes [35]. Known barriers to IBH implementation are reviewed, and we discuss how these nursing teams addressed them. Based on the experience of the authors in initiating nurse-led models and consultation roles, we suggest strategies that would improve the quality and expansion of integrated behavioral health care. To begin, what follows is a description of each model with particular emphasis on implementation issues.

## 2. Nurse Leadership in Integrated Behavioral Health Care

A review of the current literature finds eight teams of nurses that have recently published their work in developing IBH services and initiating unique CoCM models. The teams report outcomes of nurse-led IBH models or developing models of PMHNP collaboration [30,36,37,38,39,40,41,42,43] (see Table 1). In all the models, PMHNPs assumed the role of the psychiatric consultant. The models vary in terms of setting (urban vs. rural) as well as the nurse consultant role within the IBH service setting. To begin, we highlight three models demonstrating the differences in how the nurse-led model was implemented and developed. The work of these nurses is instructive as their publications trace the refinement of their IBH models.

The CoCM described in Weber et al. [42] and Stalder et al. [40] was completed in an FQHC that served all patients, regardless of payment ability. The “ACE” project, which stands for access, care coordination, and the use of evidence-based instruments, set out to improve outcomes for those with behavioral health and substance use disorders being treated in primary care. This project was funded by the HRSA Bureau of Health Workforce, Division of Nursing and Public Health (grant number, UD7HP30261). For both the adult and pediatric populations, patients entered service through primary care where behavioral health tools to measure depression, anxiety and substance use were completed. If positive, a “warm handoff ” would be initiated and the person would be screened by a behavioral health provider. Providers at the clinic included Family Nurse Practitioners (FNPs), Adult Gerontological Nurse Practitioners (AGNPs), Medical Assistants (MAs), Registered Nurses (RNs), Behavioral Health Providers (BHPs) and PMHNPs. The PMHNP was the primary provider for those with bipolar disorder, more serious substance use disorders, or those with chronic suicidality. The RN on the team assumed multiple responsibilities. For example, for individuals dealing with SMI, if indicated, the RN would facilitate their transfer to a Community Mental Health Service for more intense services.

The PMHNP was an educator and mentor for the primary care providers, especially for depressive disorders, treatment options, and co-occurring substance treatment. In addition, the PMHNP led the Medications for Opioid Use Disorder (MOUD) program so that all providers could initiate and monitor buprenorphine treatment. Outcomes for Post Traumatic Stress Disorder (PTSD), depression, and bipolar depression were significantly improved through this collaborative care model [42]. The program also supported the role of the medical and psychiatric RN practicing to the top of their scope in working as a key member of the team. The program addressed the needs of a population whose visits were funded by Medicaid, Medicare or uninsured. After the initial year, the team implemented several quality improvement teams that altered the referral process to the PMHNP psychiatric consultant and expanded the role of the PMH RN. This process is detailed below [40].

In Nebraska, in collaboration with the University of Nebraska Medical Center School of Nursing, several versions of a CoCM were developed. Cook et al. [37] report on the first year of practice of a rural satellite clinic administered by a larger FQHC which serves an agriculturally based community. This FQHC also supports a satellite clinic, which is located in a rural town of 1360 people. The satellite clinic was staffed by APRNs who provided primary care, a PMHNP consultant, and two BHPs. This nurse-run satellite clinic operated on the five core CoCM principles: patient-centered care, population-based care conducted via a registry, measured-based care, EBPs driving therapy and psychopharmacology, and the accountability of providers. Data from the first year of operation indicated significant reductions in both Patient Health Questionnaire (PHQ-9) and General Anxiety Disorder (GAD-7) scores and high levels of consumer satisfaction. This rural population was initially skeptical about their anonymity, feeling the stigma of seeking mental health care in a small town. Having mental health centered in a more “acceptable” primary care setting helped reduce this reluctance as well as enhancing the efforts of the team to build trust around protecting privacy.

Under the same grant, Emerson and colleagues [38] developed a CoCM in a larger patient-centered home. Their report covers the nine months of operation and the data collected, including both patient and organizational satisfaction with the program. Patient data demonstrated improvements in both the GAD and PHQ-9 scores. Primary care providers (PCPs)’ ratings improved around the accessibility of behavioral services and behavioral health consultation. The authors describe several strategies used to increase acceptance of the program among the PCP providers. Emerson [28] also describes the early development of a PMHNP role in another nurse-led clinic that developed a hybrid CoCM, a model so named because it retained its conceptual grounding in the CoCM principles but adapted several of the standard CoCM processes to meet the needs of their patient population. The hybrid model was conceptually grounded in the CoCM principles but differed from it in that, following a referral to the Licensed Behavioral Health Specialist (LBHS), based on acuity and need, the patient might be referred to the PMHNP, who would then meet with the patient, conduct a brief psychiatric evaluation and develop a treatment plan that the PCP would carry out. This configuration of roles reduced wait times for psychiatric appointments as well as providing tangible support for providers.

In North Carolina (NC), starting in 2013, Soltis-Jarrett drew upon a series of grants to develop a unique workforce model of integrated care. Its history was not isolated to one clinic but occurred in three areas: acute care, long-term care, and primary care settings [33] with significant success in long-term care that led to the creation of multiple PMHNP positions and the hiring of 25 PMHNP students across NC and neighboring states. Embedding PMHNPs in primary care settings proved to be challenging due to the lack of coding, billing, and reimbursement for the PMHNP services during the project timeframe (2013–2016). This led to additional funding in 2016, which set out to implement CoCM in primary care within an FQHC [30]. Using funding from HRSA (2016–2018), one PMHNP was embedded into a rural primary care setting in an attempt to implement CoCM, but it was not feasible due to the constraints of the CoCM requirements. In particular, there was insufficient staffing to meet the guidelines of the CoCM as originally planned. This led to the TANDEM3-PC model [30], which was developed and successfully implemented for two years and gathered data to ensure its success. The concepts from this unique model were then expanded to educate and train graduate NP students in a didactic traineeship and then to enhance further implementation in a post-graduate NP residency program [12,30].

Several other nurse-led models of integrated care have been developed in collaboration with Schools of Nursing. Reports are often from the perspective of the primary care NPs who were operating the primary care clinic. In Texas, a group of primary care NPs worked with the Texas A & M School of Nursing to develop integrated clinics at five sites in rural Texas [43]. They report on the first year of operation treating some 3000 patients. The collection of data to support outcomes was hampered by an EHR that was not designed for behavioral health integration, and thus it was initially difficult to document completed assessments and also hampered billing [43]. In the initial year of operation, the team also navigated COVID-19 modifications and staffing issues, including a lack of pediatric providers and insufficient funding streams for uninsured patients. Still, the team opened five clinics, showed improvements in select patient outcomes, and established work groups to improve communication and work-flow processes [43].

Similar outcomes were reported for CoCMs instituted in nurse-managed clinics serving urban, disadvantaged populations [36,39]. In these models, primary care NP providers shifted to CoCM for operations related to mental health issues and added a PMHNP to the team who acted as a psychiatric consultant (PC). Their papers describe the year of planning, how the models each scheduled visits with the care manager, and how the clinic used the PMHNP as a PC. Billing for services in both models is discussed, particularly the justification for longer visit times when a patient is seen by both the FNP and behavioral health consultant [39]. The patient data on the first year of operation reflect anticipated improvements in both depression [36] and depression/anxiety scores [39]. As with other nurse led models reviewed, these clinics addressed the needs of patients with significant economic hardship yet were able to configure workflows to address both medical and mental health needs. They also found time for significant levels of team meetings and collaboration. Their flexibility was called upon when the lack of functionality in primary care EHR required the creation of spreadsheets to manually track screening data as well as referrals and consults [36].

Another unique model, established via an academic–practice partnership and led by primary care NPs, involved the integration of behavioral health providers, a psychiatrist, a PMHNP, a clinical social worker and a care coordinator into two nurse-run clinics for uninsured and underinsured patients [41]. These clinics specialized in the treatment of patients with diabetes, via Providing Access to Healthcare (PATH) and heart failure, via Transitional Care Services for Adults (HRTSA). By establishing systems for screening and referral over the three years of operation, 520 unique patients qualified for behavioral health services (263 patients in the PATH clinic and 257 in the HRTSA Clinic). Data indicate significant reductions in depression among both patient groups; more than 60% of the patients in each group met the criteria for meaningful improvement in depression (a greater than 50% reduction in PHQ-9 score or a score less than 10). This project demonstrates the value of IBH for particularly vulnerable populations who, given their chronic illness, have traditionally high occurrences of depression. The following literature synthesis on IBH implementation situates the issues addressed by these nurse-led models.

## 3. Factors That Impact Integrated Behavioral Health Care

Evident in all these nurse-led models were the various implementation issues teams faced. These issues are not unique to nurse-led models. As IBH has been developed over the last four decades, another line of research has documented the factors related to its implementation and effectiveness [23,44,45]. Recent systematic reviews organize these barriers into categories ranging from practical considerations, such as billing and payment, to issues with role confusion and team collaboration [46,47,48]. Elaborated below are factors cited across each of these reviews. In addition, barriers are discussed related to developing IBH for vulnerable populations and those living with SMI [47,49]. We review these barriers and then discuss them in the context of nurse-led IBH models.

### 3.1. Six Commonly Sited Implementation Barriers in Integrated Behavioral Health Care

Training clinicians (including Physicians (MDs, DOs), Nurse Midwives, NPs, PAs, RNs, BHPs, and case managers to practice within integrated models is particularly relevant for effective IBH [23,47]. Practicing within an IBH setting requires both discipline-specific clinical skills and skills related to working as a team around decision-making, screening/triage, care planning, and intervention [50,51]. But several factors (e.g., ill-prepared staff, new graduates and staff motivation) pose challenges to team training [47], including siloed training of specific professions (e.g., counselors [52] or medical students [53]). Since problems in team functioning are often associated with poor implementation outcomes [54], the focus should be on enhancing team training, ideally using comprehensive guidelines (https://www.ipecollaborative.org/2021-2023-core-competencies-revision) (accessed on 5 January 2025). 

Leadership’s commitment to developing all elements of the IBH model is another factor related to effective implementation [46,47]. Poor organizational commitment often plays out via inadequate staffing, time constraints on care coordination or a focus on maximizing billable activities at the expense of staff availability for team-based care [46]. The underlying reasons for clinician lack of commitment include dissatisfaction with unclear roles and expectations as well as time demands/pressure to adequately address all aspects of IBH care [55]. Also, what may be lacking is a shared understanding of IBH or the acceptability of the model [46,47].

Issues around billing and payment appear in every list of significant barriers to integrated care expansion [46,47,48]. Financial barriers include the inability to bill for particular IBH activities such as time spent in the warm handoffs and other consultation activities [23]. Also, the fee-for-service structure is seen as inadequate reimbursement for all the patient care elements addressed within IBH [56]. Two long-standing barriers involve reimbursement for a patient receiving same-day behavioral and medical services and being unable to bill for services provided by allied providers, such as CHWs [11]. While the Center for Medicaid and Medicare (CMS) has addressed some of these issues [57], the concerns around the financial viability of IBH models continue, particularly in billing for clinician efforts to deliver all elements of patient-centered care [46].

Challenges to primary care workflow processes and demand for new communication systems are frequently identified as IBH implementation barriers [46]. For instance, fast-paced primary care workflows may not fit into the time required to address behavioral health issues [47]. While telecommunication and electronic health records can enhance workflows around collaboration, providers must still be available for consults, which can be difficult to schedule if a consultant is off-site [46]. The medical record suited for behavioral health care often does not fit well with primary care documentation, creating an additional burden on tracking IBH processes and outcomes [58].

Finally, IBH implementation has been slow to develop models that meet the needs of vulnerable populations [59]. While IBH and depression treatment have strong outcomes, the uninsured often receive depression screening in primary care but significantly fewer treatment plans or referrals [60]. One identified barrier is the lack of processes to sufficiently address the multiple co-morbidities and social needs of individuals living with SMI or dealing with substance use issues [61]. These individuals often present with complex co-morbidities, yet there is a lack of education in most MD and NP programs about the relationship of trauma, physical, and mental health, and substance use disorders [47,62]. While some IBH organizations have prioritized care for vulnerable populations [49], procedures are only slowly emerging for addressing select complexities, such as treating trauma in primary care [63].

There is little documentation of the process involved in developing single-site IBH sites, particularly from the perspective of the psychiatric provider [11]. In detailing nurse-led models and the role of PMHNPs, the reports highlighted here demonstrate strategies for dealing with the complexity of establishing roles, work processes, billing and team collaboration. Many of the reports included “lessons learned” in addressing implementation barriers. In line with the NPT framework, the strategies demonstrate how the nursing leaders adapted to context by using different approaches to utilizing staff resources and negotiating around the standard processes of CoCM models. These ideas are summarized below and organized according to the six key barriers.

### 3.2. Nurse-Led IBH Models, Implementation Barriers and Strategies to Address Training Interdisciplinary Providers

There are several levels of training needs that the nurses addressed in organizing these models. One aspect was the ongoing education of primary care providers in the care of psychiatric illness. In one setting, a Training and Implementation group was formed which identified providers and staff training needs. The PMHNP then provided both lectures and case studies on these topics [40]. In addition, educational sessions were created that focused on medication management techniques and decision-making tools to increase provider confidence in handling co-morbid mental health conditions. In another setting, a PMHNP faculty educated the team on all elements of integrated care, and then, due to excessive staff turnover, retrained and reworked aspects of the educational program for the new hires [36].

Proponents of integrated care emphasize the importance of IBH training, which does not silo practitioners into particular roles [50]. Thus, training interdisciplinary providers is best achieved with team-based training, meaning that the team trains together on the basics of integrated care and then advanced concepts for accomplishing the site’s goals. In Cook et al. [37], the team addressed the learning needs of BHPs unaccustomed to their case management responsibilities by convening all team meetings around the issue. Here, all providers learned together and worked out the specific issues with the BHP care management role. In several models, before initiating the IBH model, teams trained for nearly a year using materials from the AIMS center (Advancing Integrated Mental Health Solutions) in tandem with IBH experts to shape the resulting programs [38]. Seeing the benefits in training as a collective, and the importance of creating a common IBH language, the lead author of this report [38] recently developed a tiered, interprofessional Integrated Care Team training program to address the need to train together on common principles.

Finally, the TANDEM-3 PC takes a unique approach to training NP students and those in post-graduate training based on a model of consultation where the primary care NP and PMHNP work side by side, often seeing the patient together. Depending on the situation, either NP may take the lead in assessing the patient’s mental health needs. By working together, they learn via role modeling how to ask questions, and by dialog how to think about patient care issues [64].

### 3.3. Uneven Commitment to the IBH Model

With numerous studies documenting elements of provider dissatisfaction with IBH, it is not surprising to find varying levels of commitment to IBH among the team. Approaching this issue, Emerson [19] believes NPs should accept that particular clinicians may not have a strong buy-in to integrated care, and thus, the first step is often establishing the readiness of individual team members. PMHNP consultants assuming leadership might also encounter skepticism from primary care providers because their roles may challenge the traditional hierarchical structure of some primary care environments, where physicians have historically been designated as the leads [38].

To ensure the successful integration and effectiveness of operations in settings where physicians traditionally hold leadership roles or where providers may not be accustomed to having a psychiatric specialist guide their care, it is essential to build confidence in the abilities and competence of the PMHNP. For example, in one clinic, when the uptake of the IBH program was lower than expected, the psychiatric consultant identified and addressed barriers to utilization [38]. This process involved developing strategies to foster confidence and engagement among team members. Weekly team meetings revealed that PCPs were reluctant to implement guidance from a PMHNP given their limited interaction with this type of psychiatry provider. In recognition of this, the decision was made to conduct regular face-to-face rounds to both facilitate the opportunity to disseminate education and demonstrate PMHNP competency in an expert role. Following this practice, utilization of the program grew as PCPs saw success with the PMHNP’s recommendations [38]. Emerson [29] emphasizes that being confident in one’s sense of patients, their needs and the treatment plan is essential, as is the communication of the plan to the rest of the team, so the team interacts with patients in a consistent manner.

In both training efforts and in building commitment to their IBH model, the nurse leaders demonstrated the importance of bringing the interprofessional team together around referral processes, hand-offs, and patient care issues. Interprofessional teamwork, particularly related to sharing information within and across health and social sectors, is a critical dimension of IBH [31]. But this involves more than team meetings. As Soltis-Jarrett [30] stresses, building a team requires getting to know each team member as an individual person and coming to understand the scope of each member’s work role. Since team members work at differing levels of scope of practice and bring differing skill sets, orchestrating their efforts is critical to delivering services with a patient-centered focus.

### 3.4. Payment and Billing Issues

A prime concern in all reviews of IBH barriers is around financing; assuring payments support the reimbursement and staffing required to implement critical IBH components. This was indeed a consideration for the nurse-led clinics. One team noted the need to have personnel on staff who were experienced in billing and coding. This came to light particularly with uninsured patients. The team helped these patients enroll in the California Medicaid Program. To address this issue, they used PMHNP faculty to develop clear integrated care billing practices [36].

Noted in these nurse-led models was that in addressing financing and reimbursement, one must use resources wisely. For instance, providing mental health care to any individual who arrived at the FQHC door necessitated there be referral sources in place should the team not be able to provide the required level of care. Stadler et al. [40] emphasize that given a team and resources, one has to know who they can manage within the team and who they cannot without extensive referral. Again, the goal is to intervene early in the process before a person becomes seriously ill, and in their refined model, patients with SMI with complex needs were transferred to a mental health center. Another way to use resources wisely is for each team member to practice at the top of their scope of practice, thus not relying on PCP time for the ongoing management of patients who may need a higher level of care [40].

Most of the projects highlighted in this review were initiated via grant funding [36,37,38,39,40,41,42,43]. Thus, teams noted the need to establish a model that would be sustainable beyond grant funding. Cook et al. [37] see promise in newer CMS codes for CoCMs which will allow for the BHP and primary care NP billing of IBH services. Several authors note that their clinics were within an FQHC. This allowed for enhanced FQHC billing and support from other grants supporting the center [42]. The hybrid model allowed an organizational system where some consultation services could be billed [28] and in other situations, billing could be carried out for instances where the PMHNP provided care to complex patients [36,39]. Such diversification of funding is important. Instances in the literature detail excellent nurse-led IBH models where a reliance on philanthropy support eventually resulted in the closure of the clinic [65]. Please note that one model [39] ran successfully until 2023, when it was forced to close due to the sale of its building.

### 3.5. Developing Workflow Processes

Integrated care depends on teamwork and a good deal of communication with patient care plans. Facilitating this communication can be a challenge in a service setting where the work processes are a product of historical functioning and are predominantly driven by physicians [38]. Adapting the workflow processes to the population and the particular staffing configuration is often an ongoing task. In one setting where the referrals to the PMHNP were becoming unmanageable, a new workflow process was developed. Diagrams were created of how a patient with a particular severity level would move through the system and what providers would see the patient [40]. These diagrams were posted and facilitated understanding of roles and responsibilities and modifications to the IBH model which were implemented via aggressive use of the Rapid Cycle Quality Improvement Process. In another setting, to maintain the necessary level of support between the behavioral health team and primary care, the PMHNP used several forms of communication, phone, videoconferencing, e-consultation and on-site consults, demonstrating the flexibility needed to develop the PMHNP role and workflow processes within the system [36].

In the TANDEM3-PC model, the typical consultation ‘warm’ handoff was not part of every patient encounter. In this model, it was essential that the primary care provider was able to work to the highest level of their scope of practice and to be able to manage common psychiatric and substance use disorders. If the range of the patient’s needs was above the scope of the primary care provider, the PMHNP was then utilized to provide what the model calls “stepped-up care,” meaning providing services for more complex psychiatric conditions that would traditionally be beyond a CoCM model. This process continues to evolve. Currently, with the possibility of hiring a PMHNP available, the primary care team will be able to refer stepped-up care for patients (followed by the PMHNP) until the patient is stable and/or in remission. This strategy also promotes the opportunity for the primary care provider to continue to refer challenging cases to the PMHNP without the delay from outside referral agencies.

Thus, as depicted in the NPT organizational framework, the nurse-led models often employed unique strategies to fit the service site context while retaining elements of an IBH model. Using the hybrid CoCM approach enabled the customization of clinic site processes which leveraged its inherent strengths while developing a sound understanding of the clinical challenges that might affect implementation [29]. Additionally, hybrid models acknowledge clinics may have various degrees of integration already occurring which can be built upon rather than uprooting and changing work flows completely. For instance, clinics that employ PCMH inherently have processes in place to facilitate the integration of services but may lack some of the systematic measurement-based strategies or tracking mechanisms seen in Hybrid or CoCMs. It would only make sense then to fully understand what work is currently being carried out and use that knowledge as a foundation to build upon versus ceasing those activities and implementing a different process.

### 3.6. Addressing the Needs of Complex Populations

As explained, knowing what type of patient your team and resources can effectively treat is essential. In one model, to mitigate the referral of all patients with mental health issues to the PMHNP, a tier system was formed based on patient complexity [40]. Therefore, patients with a fairly straightforward mild depression or anxiety were treated by the PCP. As a patient’s complexity increased (Tier 3 and 4), their needs were addressed by a wider variety of team members. For instance, Tier 3 patients who might be dealing with bipolar disorder, treatment-resistant depression or co-morbid trauma were seen directly by the PMHNP [40]. To clarify the roles and responsibilities for each patient tier, a process map was developed to demonstrate the patients’ typical movements through the system. One outcome measure of the system was assessing if patients were assigned to the appropriate tier and, within that, if staff were functioning to the full extent of their scope. Data indicated that as the year progressed, referrals to the psychiatric team decreased, indicating more patients were being seen by the PCP [40].

Another strategy for handling complex patients involved the optimal use of the treatment team, particularly the RN case manager. In the tiered system mentioned earlier, any patient above tier one was referred to the RN and PMHNP, who collaborated to assess acuity and needs. The RN also monitored ongoing care through an AIMS tracker and coordinated services [40]. Nurse case management is a well-supported approach, along with task shifting, where nurses lead groups and provide low-intensity mental health interventions [66]. Optimizing RN utilization demands a common understanding of all team members’ scope of practice and to employ RNs in patient support, education, and outreach. Another tool to engage individuals with complex care needs is the use of peer support/CHWs, but one needs to be clear about their scope of practice and reimbursement guidelines [11].

## 4. Discussion

This report traces the implementation and development of IBH/CoCM by eight nurse-led teams. The clinic-based projects posted positive outcomes for patients, particularly in the areas of depression and anxiety. The IBH clinics addressed the needs of underserved populations in both rural and urban areas that were largely uninsured and underinsured, and, in one instance, the needs of patients with chronic disease. While development and some of the hiring were dependent on grant funding, the authors anticipate their models being sustained owing to their relationships with the FQHCs and Academic Health Centers or enhanced billing codes.

All teams encountered difficulties initiating a new service model, from establishing new workflows and team communications to augmenting EHR capabilities to track data. They also encountered unique circumstances during COVID-19, which brought about a significant staff turnover. Often, teams needed to problem-solve for the underutilization or conversely the overuse of PMHNP consultation, using multiple forms of communication to establish PMHNP collaborations. In some instances, new workflows were created and modifications of standard CoCM processes were made. Each solution required the nurses’ flexibility, problem-solving, and systems knowledge. In line with the NPT model, IBH models are context-sensitive and replicating them demands understanding the organizational culture of the broader system and how IBH methods and values align with that culture. The nurse-led models demonstrated how one adopts the IBH model to specific contexts and patient needs. Thus, to replicate any nurse-led model will require translating IBH principles to the designated setting and focusing on what works in that setting with that specific population.

Their models point to several strategies for enhancing and expanding IBH. Lessons learned in these models inform us about factors which would assist in scaling up nurse- led IBH models. One strategy for scaling up these IBH models involved utilizing each health care team member to the top of their scope of practice. Several models demonstrated the optimization of the RN role. In one model, the nurse assumed the case manager role and helped the patient navigate within the system and into the community. In another, the RN worked with the PMHNP to determine the level of severity and placement of patients in the site’s tier system. The literature supports the expanded role of RNs within IBH in care management, to assist patients with self-management and through the delivery of low intensity mental health interventions [66,67]. Innovative programs to train nursing students in integrated care are appearing in the literature and demonstrating positive outcomes in student communication, leadership and critical reasoning [68]. Such programs should be replicated to develop a cadre of RNs capable of practicing in IBH and extending the capacity of the care team. Recognizing that recent PMHNP graduates may need additional training to work within integrated care, the expansion of IBH might include post-graduate education as well as a continuation of educational offerings [23], particularly as a strategy to increase PMHNPs in rural areas [69].

To sustain IBH models, patients must attend appointments and engage in their care. The projects also point to another important area to enhance IBH reach. As noted by one team, some patients at both clinics were not engaged, declined services or never attended their BH care appointments [41]. Another team discussed how stigma combined with its social consequences challenged engaging rural patients in integrated services [37]. Poor treatment engagement in integrated care is complex, related to both traditional skepticism around mental health services [70] and factors related to the coordination of IBH services, including poor understanding of IBH and misalignment of system priorities with patient goals [1]. As illustrated by several models, developing trust in the providers, the system and its care is essential [71]. Recent scholarship on treatment engagement recommends a paradigm shift that moves away from strategies aimed at individuals and instead enlists the individual, family and community as critical co-designers of integrated care research and decision-making [72]. As NPs assume leadership roles in IBH, they should consider how to draw upon consumer input to build IBH systems that fit within the context of the community.

To sustain federal support and outside funding, IBH models must demonstrate outcomes [59]. One team [39] suggested determining IBH effectiveness by broadening data collection on outcomes to include patient satisfaction, team knowledge and satisfaction with the model, factors that would be measured at regular intervals throughout implementation. As IBH is a team effort, it would also make sense to evaluate outcomes that reflect team-based functioning [22]. A recent report of a nurse-managed clinic with both a chronic disease and behavioral health management goal assessed efforts to improve and evaluate team communications [3]. Their report demonstrates how one might consider team communication to determine IBH effectiveness, particularly acknowledging the inclusion of allied health professionals. Nurse-led model are often created in the context of academic–practice partnerships. Such partnerships form a team with faculty members who can assist with the development and implementation of evaluation plans that could build a more comprehensive sense of IBH effectiveness and team functioning [58].

### Limitations

This report focused on recent CoCM models that used PMHNPs as the psychiatric consultant. Early groundbreaking models of nurse-led behavioral health integration were not discussed since much has changed in the intervening years [73,74]. We did not include nurse-managed clinics that have developed integrated care with a team of psychiatric providers, e.g., [75], or nurse- managed clinics that address a wide range of health care including mental health needs [76]. Issues of financial viability were not considered as this requires in-depth analysis of billing and revenues [77]. Concerns surrounding integrated care for life span populations were not reviewed, but both children and older adults have significant mental health needs and would benefit from expanding integrated behavioral health care services [78,79,80].

## 5. Conclusions

The importance of IBH and nursing leadership in developing these models has been endorsed by both of the major professional US psychiatric nursing organizations [81,82,83]. The PMH workforce at both the RN and NP levels have the capacity and the capabilities to fill critical roles in mental health service delivery, including the IBH model [28,84,85,86,87,88,89] and the stronger integration of substance use services within these models [90]. The nurse-led models highlighted here demonstrate their focus on developing IBH that met the needs of the underserved and rural populations. Given the significant number of PMHNPs in the behavioral health workforce, we anticipate their continued involvement in the development of effective IBH models.

### Future Directions and Policy Recommendations

1. We need to promote the development of nurse-led models of IBH with particular emphasis on services tailored to underserved and rural locations. This may require incentives for PMHNPs to practice in these areas. Policies on loan repayment that favor student loan repayment for NPs who practice in rural/underserved areas should expand the criteria for qualified employment sites. HRSA support should continue the support for NP residencies conducted in rural and underserved sites which influences NP decisions to work in these areas [69].

2. Research demonstrates the importance of providers having experiences in integrated care during training. Educators should increase interdisciplinary training for IBH via placements in community-based settings where students have hands-on experience with communities and the lives of patients [68,91], particularly in rural setting [92]. The Bipartisan Policy commission report on strengthening the IBH workforce recommended that HRSA leverage the Title VII and VIII training programs and the Teaching Health Center program to augment real-world integrated care experiences for the future health care workforce [22].

3. When the PMHNP functions as a psychiatric consultant in CoCM models, most federal and commercial insurance allow for billing for services where the PMHNP intervenes directly with high-need or complex patients. CMS regularly provides guidance on the Behavioral Health Integration and Collaborative Care billing codes [57]; however, the use of these codes can be confusing, particularly when specifying time spent in collaborative care activities vs. billing for the provision of direct psychiatric services [58]. CMS should continue to clarify the use of these codes and gather data on their utilization and impact on quality.

4. CMS allows for same-day billing by primary care and specialty psychiatric providers if services are distinct and medically necessary; however, this varies with specific state regulations [93]. Integrated care providers and organizations should advocate for billing codes and reimbursement rates that match the IBH services provided with consistency across the states.

## Figures and Tables

**Table 1 ijerph-22-00598-t001:** Summary of nurse-led models, title, setting, challenges addressed.

Authors	Focus	Site	Key Features	Challenges/Adaptions
Brich et al., 2021 [36]	Depression and physical health metrics	Urban FQHC in area of high poverty	University faculty partnered with team for education and team/clinical process development	Addressing needs of patients with significant medical issues, poor EHR functionality/configuring workflows to address both medical and mental health needs data, developing spreadsheet software to manually track screenings and referrals
Cook et al., 2024 [37]	Depression and anxiety	Rural satellite clinic of an FQHC	Operated on all five CoCM principles; situated to serve a rural population in a small town	Addressing the perceived stigma of seeking mental health care and threats to anonymity/focused efforts of the team to build trust around protecting privacy
Emerson et al., 2023 [38]	Depression and anxiety, provider satisfaction	Urban academic primary care clinic	Clinic developed via training grant for both NP training and development of integrated services	Clinic culture steeped in top–down processes for non-integrated primary care/instituted weekly team meetings to build confidence in PMHNP consultation
Reising et al., 2023 [39]	Depression and anxiety	Urban FQHC in disadvantaged area of City	Clinic developed to serve populations with significant socioeconomic barriers to health care	Need to adjust clinic schedule to allow for longer client visits and also allow staff time for client case consultation/close collaboration of primary care (PC) providers and behavioral health staff helped build PC providers’ confidence in addressing behavioral health issues
Stadler et al., 2023 [40] Weber et al., 2021 [42]	Depression, anxiety and substance use services for all patients regardless of ability to pay	Nurse-managed FQHC in urban areas that also operates as Faculty Practice site	RN on team as case manager for individuals dealing with Serious Mental Illness; embedded MOUD program	Initial referral process called for all patients in need of medication management to be referred to PMHNP/expanded role of RN; re designed a referral program to the PMHNP for psychiatric consultation
Talley et al., 2021 [41]	Depression, anxiety and several health markers	Nurse-run clinics for uninsured patients specializing in Diabetes and Heart failure	Addressing co-morbid depression in an IBH clinic designed for vulnerable populations with chronic illness	Establishing systems for screening and referral within the clinic process; patient engagement in services/designed processes for completing screening; conducted detailed analysis of patient factors related to engagement
Soltis-Jarrett, 2019 [30] Soltis-Jarrett, 2016 [32]	Depression and development of NP training	Rural primary care clinics	Over 6 years, integrated care training model developed that facilitated skill development of primary care NPs and PMHNP	Clinic not able to meet the staffing requirements of CoCM model/development of a unique training model for NP students and post-graduate NP residency program
Weston et al., 2023 [43]	Depression and medical markers	Rural health clinics in five sites in Rural Texas	Serving patients many of whom were living below poverty level and were uninsured.	Billing and outcome data collection impeded by EHR not designed for IBH, high staffing turnovers/established work group to improve communication and workflow

## Data Availability

No new data generated.

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
