# Peer review of "Engineering Integrated Care Expansion and Innovation: Drawing upon Nursing Leadership"

_ijerph, 2025, doi:10.3390/ijerph22040598_

Round 1
Reviewer 1 Report
Comments and Suggestions for Authors
This paper provides valuable insights into the expansion of nurse-led Integrated Behavioral Healthcare (IBH) models; however, several areas could be strengthened to enhance clarity, methodological rigor, and practical applicability. Below are key recommendations for improving the overall quality of the manuscript
- The introduction provides a strong background on Integrated Behavioral Healthcare (IBH), but it could benefit from a clearer statement of the research gap and the paper's specific contribution.
- Some sections shift topics abruptly. Consider adding transition sentences between major sections to improve readability.
- The paper discusses eight nurse-led IBH models but lacks a comparative analysis. Consider including a table comparing key characteristics, outcomes, and challenges of these models.
- Some claims about IBH effectiveness and PMHNP roles are supported by citations, but there are instances where additional empirical evidence or references would improve credibility (e.g., claims about workforce shortages and training gaps).
- Some terms, such as "hybrid models" and "stepped-up care," are used without clear definitions. Providing definitions or brief explanations would improve reader understanding.
- The paper mentions financial barriers and reimbursement challenges but does not explore specific policy recommendations for overcoming these issues.
- While the study discusses successful nurse-led IBH initiatives, it does not sufficiently address how these models can be replicated or scaled in different healthcare settings.
Author Response
Thank you very much for taking the time to review this manuscript. Please find the detailed responses below and the corresponding revisions/corrections highlighted/in track changes in the re-submitted files
The introduction could benefit from a clearer statement of the research gap and the paper's specific contribution. |
Agree, added a statement on research gap: lack of research synthesis around nurse-led models of collaborative care (lines 99-102). |
Some sections shift topics abruptly. Consider adding transition sentences between major sections to improve readability. |
Transition sentences added between major sections (lines 124, 247, 317). |
Consider including a table comparing key characteristics, outcomes, and challenges of these models. |
Added a table on models comparing sites, characteristics and challenges (lines 135-136) |
Some claims about IBH effectiveness and PMHNP roles are supported by citations, but there are instances where additional empirical evidence or references would improve credibility (e.g., claims about workforce shortages and training gaps). |
These are indeed issue (workforce shortages and training gaps) that have support in literature. Evidence on these points, with references have been added. (lines 72-74; lines 92-95) |
Some terms, such as "hybrid models" and "stepped-up care," are used without clear definitions. Providing definitions or brief explanations would improve reader understanding. |
Thanks for pointing this out. Definitions provided for these terms. Agree it will improve understanding of the broad concept – for hybrid model (line 186); stepped care ( line 443). |
The paper mentions financial barriers and reimbursement challenges but does not explore specific policy recommendations for overcoming these issues. |
Policy recommendations around financial barriers (mainly around reimbursement and billing) have now been added to the final section of the paper. (lines 568- 592) |
While the study discusses successful nurse-led IBH initiatives, it does not sufficiently address how these models can be replicated or scaled in different healthcare settings. |
Thoughts added on lessons learned from these models about how to adopt IBH to a particular context. Also made clear in discussion that the three strategies detailed are directly related to scaling up models. (lines 506-547). |
Reviewer 2 Report
Comments and Suggestions for Authors
Manuscript ID ijerph-3505149 Review reports
Comments and Suggestions
- Summary:
The manuscript, titled "Engineering Integrated Care Expansion and Innovation: Drawing upon Nursing Leadership," explores the role of Psychiatric Mental Health Nurse Practitioners (PMHNPs) in leading integrated behavioral healthcare (IBH) models. The paper discusses eight nurse-led IBH models, the implementation challenges they faced, and strategies for overcoming these barriers. The study provides valuable insights into workforce development, team-based care, and the expansion of IBH services, particularly for underserved populations. The paper concludes by advocating for policy changes, increased training, and financial support to sustain and scale these models.
- General Comments:
This manuscript presents a timely and well-researched discussion on the role of nursing leadership in integrated behavioral healthcare (IBH). By examining eight nurse-led IBH models, the study provides valuable insights into workforce development, implementation strategies, and healthcare service expansion. The paper effectively highlights the critical role of Psychiatric Mental Health Nurse Practitioners (PMHNPs) in leading IBH initiatives and overcoming system-level barriers.
One of the manuscript’s greatest strengths is its practical relevance. The discussion is well-grounded in current healthcare challenges, particularly the need for interdisciplinary collaboration, policy support, and sustainable workforce strategies. The study is also highly applicable to policymakers, healthcare administrators, and nursing educators, offering concrete recommendations for expanding IBH services.
While the manuscript is strong in content and contribution, there are areas where it could be further enhanced. Providing greater theoretical grounding, deeper engagement with the literature, and a more structured presentation of findings would improve the rigor and impact of the paper. Additionally, a clearer articulation of the methodology used to select and analyze the IBH models would enhance the transparency and reproducibility of the study.
- Specific Comments:
(1) Theoretical Framework:
The paper presents a compelling discussion of nurse-led IBH models, but it would benefit from a more explicit theoretical framework. The discussion on barriers and facilitators of IBH implementation would be strengthened by linking them to established healthcare implementation frameworks.
(2) Literature Review:
The literature review is well-researched and comprehensive, covering key topics related to integrated care, nurse leadership, and behavioral health models. The paper effectively highlights the gaps in existing research, emphasizing the need for further exploration of nurse-led IBH initiatives. One possible improvement would be to increase critical engagement with the reviewed studies. Rather than summarizing key findings, synthesizing the literature to identify common themes and trends would strengthen the discussion.
(3) Methodology:
The manuscript presents a well-structured analysis of eight nurse-led IBH models, offering rich insights into their implementation, challenges, and successes. Clarifying the selection criteria for these models would enhance the transparency of the study. Were these models systematically identified, expert-recommended, or selected based on specific characteristics?
(4) Findings:
The findings are highly informative and relevant, offering practical insights into the implementation of nurse-led IBH models. The discussion of workforce challenges, financial barriers, and team-based care strategies is particularly valuable. Addressing potential scalability and sustainability challenges of nurse-led IBH models would enhance the study’s policy relevance.
- Questions and Suggestions:
(1) How were the eight IBH models selected? Providing more details on the selection process would enhance methodological clarity.
(2) How do nurse-led IBH teams collaborate with other healthcare professionals? A brief discussion on interprofessional teamwork would add depth to the findings.
- Overall:
This manuscript makes an important contribution to the literature on integrated behavioral healthcare and nursing leadership. It is well-structured, informative, and offers practical implications for policymakers and healthcare leaders. The discussion is highly relevant, addressing critical issues in workforce development, financial sustainability, and interdisciplinary collaboration. The paper is well-written and valuable. To further strengthen the manuscript, the authors should consider:
(1) Structuring the findings under key themes for better readability.
(2) Expanding the discussion on policy and scalability to emphasize real-world implementation strategies.
Author Response
Thank you very much for taking the time to review this manuscript and your encouraging comments. Please find the detailed responses below and the corresponding revisions/corrections highlighted/in track changes in the re-submitted files
Questions and Suggestions |
Response |
How were the eight IBH models selected? Providing more details on the selection process would enhance methodological clarity. |
These details have been added – it was an unstructured search but specific criteria were applied to the selection of models (lines 106-113) |
How do nurse-led IBH teams collaborate with other healthcare professionals? A brief discussion on interprofessional teamwork would add depth to the findings. |
In all of the models one sees how the nurse leader built the team focus on both processes and patient care issues. A paragraph was added to emphasize these strategies and emphasize the importance of interprofessional team (line 378-386) |
Overall: Structuring the findings under key themes for better readability. |
The key findings are organized into two sections, a description of the models and then the challenges faced and the adoption of processes to particular contexts/patient populations. This is in line with the organizational model that is now explicitly named. We have highlighted the paragraph that explains this emphasis on synthesizing findings. (line 313-316) |
Expanding the discussion on policy and potential scalability and sustainability to emphasize real-world implementation strategies. |
We have added a paragraph on two principles of scalability and then added several policy recommendations to support nurse-led models (lines 500-504) (lines 567-591) |
Additional commentary |
|
Paper would benefit from a more explicit theoretical framework. The discussion on barriers and facilitators of IBH implementation would be strengthened by linking them to established healthcare implementation frameworks. |
We have added a brief discussion of Normalization Process Theory which works well with how the models were implemented and the influence of context- which demands specific mechanisms of resource mobilization, collective action and negotiations with context. (lines 115-120) The NPT idea of adopting to context is treaded through the paper. |
.
Reviewer 3 Report
Comments and Suggestions for Authors
Dear authors, congratulations on the work you have developed.
The article discusses the expansion and innovation of integrated mental health care in the United States, focusing on nursing leadership and the role of Psychiatric Mental Health Nurse Practitioners (PMHNPs) in nurse-led models. This is a highly relevant topic that addresses a critical public health issue—the need to expand access to mental health care for particularly vulnerable populations.
However, I believe there is space for some improvements, namely:
- Improving clarity and conciseness; for example, the section on training could be revised to avoid repetitions and reduce redundancies.
- Although financial challenges and sustainability strategies are addressed, a more in-depth analysis of the financial viability of the models could strengthen the argument.
- The conclusion section could be more concise and more clearly highlight the practical implications of the study for future implementation.
Author Response
Thank you very much for taking the time to review this manuscript and your encouraging comments. Please find the detailed responses below and the corresponding revisions/corrections highlighted/in track changes in the re-submitted files
Improving clarity and conciseness; for example, the section on training could be revised to avoid repetitions and reduce redundancies.
|
This section has been edited to remove redundancies. We attempted to balance the length of each section and omitted what might be considered extraneous sentences throughout the paper. Training sections were shortened (lines 89-97 & lines 260-274). |
Although financial challenges and sustainability strategies are addressed, a more in-depth analysis of the financial viability of the models could strengthen the argument. |
As noted in the limitations section, more data on billing, staffing and sources of funding would be needed to analyze the financial viability of the models. We just did not have the data to broaden that discussion. In policy recommendations we discuss the need to clarify billing regulations. (lines 582-592). |
The conclusion section could be more concise and more clearly highlight the practical implications of the study for future implementation. |
Conclusion tightened a bit. We did clearly identify the three practical implications for future implementation (drawn from the study. Lines 506 -557) . |
Round 2
Reviewer 3 Report
Comments and Suggestions for Authors
Congratulations!